# Finding agreement in disagreement: Simultaneous label alignment and multi-dataset training with SLAMDUNKS

## Abstract

Multi-dataset training is a key strategy for improving the versatility and robustness of deep models, but its effectiveness is often hindered by unaligned and contradictory dataset taxonomies. These inconsistencies introduce training noise and prevent effective knowledge sharing. To address this, we propose SLAMDUNKS, a framework for simultaneous multi-dataset training and label alignment. Its core is a shared feature extractor trained with two competing heads: a gating head that determines which dataset-specific classes should be shared, and a classification head that maps samples to the emerging shared taxonomy. To rigorously evaluate alignment quality, we introduce a synthetic benchmark where ground-truth relations are modeled as bipartite graphs. Our method demonstrates remarkable precision, perfectly recovering the true taxonomy (a Graph Edit Distance of 0) for same-domain datasets. Across more challenging cross-domain pairs, SLAM-DUNKS achieves an Average Precision of 0.8, outperforming the state-of-the-art by 0.1 to 0.2 and validating its superior alignment capabilities.

## 1 Introduction

Machine learning has driven remarkable progress in domains such as natural language processing DeepSeek-AI (2025), medicine Schmidt et al. (2024), and climate science Eyring et al. (2024). A key driver of this success is labeled data, which connects raw samples to meaningful applications. However, even within the same domain, datasets often reflect different labeling perspectives. For example, an animal can be classified by species, diet, or reproductive strategy. A platypus may therefore be grouped with mammals (e.g., lions), carnivores (e.g., venus flytraps), or egg-laying species (e.g., birds). This diversity produces specialized datasets with overlapping content that could, in principle, be combined for better training Zhou et al. (2022); Bevandić et al. (2024). Yet combining datasets requires mapping samples into a common taxonomy, a process that is rarely straightforward: labeling policies may be inherited Krešo et al. (2018), redesigned Lambert et al. (2020), or left ambiguous, and automation remains difficult.

Vision–language models offer partial solutions by aligning classes through names or descriptions Li et al. (2022), or dynamically relabeling into new taxonomies with foundation models Sun et al. (2025). However, these approaches often require human intervention or rely on manually defined taxonomies, and purely language-based alignment is prone to inconsistencies. For example, in COCO Lin et al. (2014), tie is a separate class, while in ADE20K Zhou et al. (2017) it belongs to person Bevandić & Šegvić (2022). Both datasets also include carpet and floor, but differ in how carpeted floors are annotated Uijlings et al. (2022). Even recent open-vocabulary methods employ ad hoc class renamings Ghiasi et al. (2022); Xu et al. (2023), which may improve performance but are rarely systematic or sufficiently discussed. Such inconsistencies undermine rigorous evaluation by obscuring the semantic relation between training and test data.

Language can nonetheless provide a useful signal for discovering class relations, but models must also observe dataset samples directly. Prior work shows that unified taxonomies can be constructed if class mappings are known Bevandić et al. (2022), or recovered when relationships are specified Bevandić & Šegvić (2022). We consider two classes related if they share at least one underlying

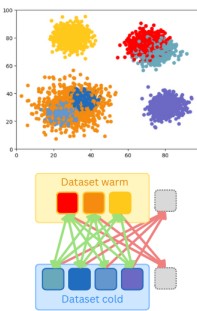 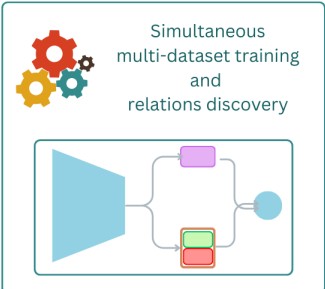 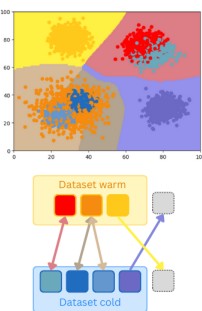

Figure 1: Overview of our proposed approach. Given a dataset pair, we start from the assumption that all the classes may be related to each other or to an additional rest of the world class. Our model has two competing heads on top of a shared feature extractor. Our custom loss results in a model which finds the correct semantic relations and is able to reason into a recovered shared taxonomy.

concept—for instance, mammals and egg-laying species are linked by the platypus, and the shared concept of an egg-laying mammal can itself be treated as a standalone class.

We propose SLAMDUNKS (Synchronising Labels Across Multiple Datasets for UNinterrupted Knowledge Sharing), a model that simultaneously discovers semantic relations and trains a classifier in the unified taxonomy. Starting with two datasets with only dataset-specific labels, we assume each class may relate to any class in the other dataset or remain standalone. Each potential relation defines a candidate in the universal taxonomy, establishing an upper bound on trainable categories. Training proceeds over all candidates, and only valid concepts remain active. The resulting model classifies samples into the unified taxonomy while revealing class relations (Figure 1).

Evaluating such methods is equally important. Standard metrics based on single-dataset performance Lambert et al. (2020); Bevandić & Šegvić (2022) can be misleading, as improvements may simply reflect model capacity rather than genuine alignment Rong et al. (2024). To address this, we propose an evaluation framework with controlled ground-truth relations. By constructing custom taxonomies from existing datasets, we test alignment quality directly. We measure relation discovery with average precision (AP) and graph edit distance (GED), and report accuracy on the unified taxonomy to assess multi-dataset training.

Our contributions are as follows:

- We formalize relation discovery as a standalone task, introducing synthetic dataset collections with controllable ground-truth relations and evaluation via AP and GED.
- We present SLAMDUNKS, a model that simultaneously aligns taxonomies and trains classifiers across datasets.
- We validate SLAMDUNKS extensively, showing that it outperforms existing methods and uniquely handles standalone classes.

## 2 RELATED WORK

We discuss the two tasks performed by SLAMDUNKS: multi-dataset training and semantic relation discovery in dataset collections.

### 2.1 MULTI-DATASET TRAINING

Multi-dataset training continues to be a powerful paradigm for improving generalization and mitigating dataset-specific biases. Initial strategies either used dataset-specific prediction heads on shared backbones Kalluri et al. (2019), or merged datasets without resolving label inconsistencies Masaki et al. (2021). Subsequent refinement introduced cross-logit interaction Fourure et al. (2017) and open-set recognition techniques to detect unknown or novel classes Chan et al. (2021); Biase et al. (2021); Uhlemeyer et al. (2022), treating non-primary datasets as anomaly or negative domains Tian et al. (2021).

Label taxonomy alignment has also evolved. Approaches using hierarchical structures Liang et al. (2018); Meletis & Dubbelman (2018) often required complex pipelines, while manual relabeling collapsed labels at the expense of specificity Lambert et al. (2020); Zendel et al. (2022). A more scalable approach constructs a universal taxonomy, representing dataset-specific labels as unions of universal classes—supporting consistent multi-dataset training without altering annotations Bevandić et al. (2024); Cour et al. (2011).

Recent works have further advanced the field: DaTaSeg Gu et al. (2023) and LMSeg Zhou et al. (2023) leverage modular architectures and language-driven objectives for cross-dataset alignment. LMSeg, in particular, employs textual supervision (e.g., CLIP) to link heterogeneous annotations dynamically. HTTS Meletis & Dubbelman (2023) integrates a taxonomy-aware transformer architecture, exploiting hierarchical label priors via network design. MultiTalent Ulrich et al. (2023) demonstrates improved results by co-training across varied medical segmentation datasets. TMT-VIS Zheng et al. (2023) applies taxonomy-aware joint training within video instance segmentation. Plain-Det Shi et al. (2024) delivers competitive performance through architectural simplicity and carefully designed multi-dataset training strategies.

## 2.2 AUTOMATED CROSS-DATASET CLASS RELATION DISCOVERY

Manual label alignment becomes increasingly impractical at scale. Semantic text embeddings (e.g., CLIP-based) offer automation Li et al. (2022); Yin et al. (2022), but are unreliable due to label ambiguity Uijlings et al. (2022). Visual cues, in contrast, often yield more accurate alignments.

One line of work uses post-hoc optimization (e.g., linear programming) on output logits to match classes, yet these methods assume one-to-one class correspondence and fail to handle hierarchical relations—leading to competition and reduced performance Zhou et al. (2022); Bevandić et al. (2022).

Newer, more flexible approaches include: DynAlign Sun et al. (2025) performs dynamic, unsupervised alignment during training using foundation model priors—allowing detection of fine-grained and overlapping categories via joint optimization HTTS Meletis & Dubbelman (2023) enforces hierarchical consistency through its taxonomy-aware transformer architecture. AutoUniSeg Rong et al. (2024) employs graph neural networks to infer cross-dataset class relations by jointly considering semantic and visual feature spaces. RESI Zhangli et al. (2024) targets inconsistencies in label semantics through structured, task-specific label alignment.

Variants of language-vision approaches continue to emerge. LMSeg Zhou et al. (2023) integrates text-based label representations into training, replacing hard labels with text-guided supervision and enabling seamless adaptation across datasets and tasks.

Our work differentiates itself by embedding taxonomy discovery directly into the training loop, jointly optimizing segmentation performance and label alignment within a universal taxonomy. This unified, data-driven process supports taxonomic overlap, label granularity variation, and semantic inconsistency—without relying on manual mappings or fixed label embeddings.

## 3 METHOD

We introduce **SLAMDUNKS**, a model designed for simultaneous multi-dataset training and semantic relation discovery between class taxonomies. The key idea is to embed the discovery of cross-dataset relations directly into the training loop. After training, the model can classify objects according to the induced universal taxonomy, without requiring any additional post-processing or manual label mappings.

### 3.1 PRELIMINARIES

We assume that two classes are related if they share at least one common visual concept. For example, the classes *Vistas-car* and *ADE-van* are related because both include pickup trucks.

We represent such inter-dataset class relations in two equivalent ways:

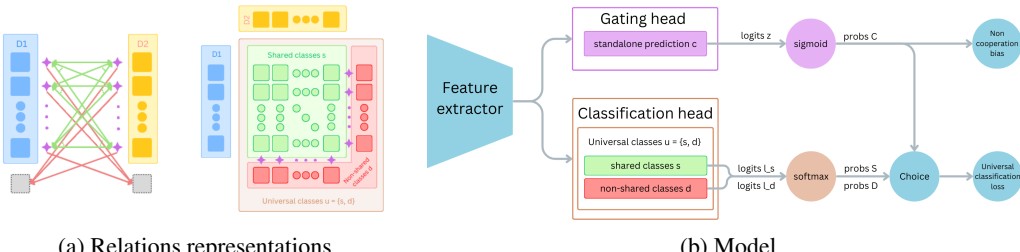

(a) Relations representations                    (b) Model

Figure 2: Method

1. **Bipartite Graph.** Vertices correspond to dataset-specific classes, and we introduce an additional dummy vertex for each dataset to represent the *rest of the world (RoW)*—concepts outside the given taxonomy. Edges connect related classes. If a class has no valid relation to the other dataset, it connects to its dataset's RoW vertex, marking it as a *standalone class*.

2. **Universal Class Decomposition.** Alternatively, we can define a set of primitive visual concepts (universal classes) that appear across datasets, and express dataset-specific classes as unions of these universal classes. The two views are interchangeable, since each bipartite edge encodes a potential universal class.

When relating two datasets $D_1$ and $D_2$, we make no assumptions about prior alignments. Initially, any class in $D_1$ may be related to any class in $D_2$—including the dummy RoW class. Multiple relations per class are permitted, except for RoW connections, which indicate that the class is entirely standalone. This setup defines an upper bound on the number of recoverable universal classes:

$$(|D_1| + 1)(|D_2| + 1) - 1.$$

Figure 2a illustrates this framework. Green edges denote potential inter-class relations, red edges connect classes to the RoW vertex (standalone classes), and purple gates indicate the model's choice between shared and standalone assignments. The bipartite graph can also be represented as a matrix of universal classes, where each dataset-specific class corresponds to a union of rows or columns in the matrix.

### 3.2 SLAMDUNKS

Our model builds on a shared feature extractor with two task-specific heads that jointly determine class relations and classify samples in the universal taxonomy.

**Gating Head.** The first head produces $|D_1| + |D_2|$ logits, denoted by $z$, with a sigmoid applied to each. This head can be interpreted as a collection of binary classifiers, one per dataset-specific class, that predicts whether the class is *shared* with the other dataset or *standalone*. These predictions correspond to the purple gates in Figure 2a.

$$P(C_y = 1 \mid \boldsymbol{x}) = \frac{1}{1 + \exp - z_y}. \tag{1}$$

**Classification Head.** The second head outputs $(|D_1| + 1)(|D_2| + 1) - 1$ logits, followed by a softmax. This head assigns each sample to one of the universal classes, covering all theoretically possible relations. Specifically: - $|D_1||D_2|$ logits $l_s$ represent all candidate *shared classes*, and - $|D_1| + |D_2|$ logits $l_d$ represent all possible *standalone classes*.

$$P(S = s' \mid \boldsymbol{x}) = \frac{\exp l_{s'}}{\sum_{s \in \boldsymbol{S}} \exp l_s + \sum_{d \in \boldsymbol{D}} \exp l_d}. \tag{2}$$

$$P(D = d_y \mid \boldsymbol{x}) = \frac{\exp l_{d_y}}{\sum_{s \in \boldsymbol{S}} \exp l_s + \sum_{d \in \boldsymbol{D}} \exp l_d}. \tag{3}$$

## 3.3 TRAINING OBJECTIVE

The two heads of SLAMDUNKS interact to define the posterior probability of each dataset-specific class. Intuitively, the cooperation (gating) head decides whether a class is *shared* or *standalone*, while the classification head provides the distribution over all candidate universal classes.

**Shared classes.** If a dataset-specific class $y$ is shared, its posterior probability equals the product of (i) the probability that it is not standalone, and (ii) the probability mass assigned to the set of related universal classes:

$$P(Y = y \mid x, s) = P(C_y = 0 \mid x) \sum_{s' \in m_{S_d}(y)} P(S = s' \mid x). \tag{4}$$

**Standalone classes.** If $y$ is not shared, the posterior is given by the product of (i) the probability that it is standalone, and (ii) the probability of its associated standalone universal class:

$$P(Y = y \mid x, ns) = P(C_y = 1 \mid x) P(D = d_y \mid x). \tag{5}$$

The two remaining combinations of outputs are invalid and ignored.

**Total posterior.** The overall posterior probability of class $y$ is obtained by summing the two valid cases:

$$P(Y = y \mid x) = P(C_y = 0 \mid x) \sum_{s' \in m_{S_d}(y)} P(S = s' \mid x) + P(C_y = 1 \mid x) P(D = d_y \mid x). \tag{6}$$

Expanding this expression highlights the role of the cooperation logits $z$:

$$P(Y = y \mid x) = \frac{\sum_{s' \in m_{S_d}(y)} \exp(l_{s'} - z_y) + \exp(l_{d_y})}{(1 + \exp(-z_y)) \left( \sum_{s \in S} \exp(l_s) + \sum_{d \in D} \exp(l_d) \right)}. \tag{7}$$

This formulation shows that $z_y$ dampens shared logits $l_s$ whenever the model favors a standalone interpretation, allowing the standalone logit $l_{d_y}$ to dominate. Without this mechanism, standalone logits would not be learnable, as they always co-occur with shared ones—an issue observed in weakly supervised settings Cour et al. (2011); Bevandić et al. (2024).

**Loss function.** We train SLAMDUNKS by minimizing the negative log-likelihood of the dataset-specific posterior, augmented with a bias toward the standalone prediction:

$$\mathcal{L}(x, y \mid m_{S_d}) = -\ln P(Y = y \mid x) - \lambda \ln P(C_y = 1 \mid x). \tag{8}$$

This objective creates a competition between the two heads: the cooperation head pushes classes toward standalone assignments, while the classification head pulls them toward shared relations when evidence supports it. If two classes are easily distinguishable, the bias toward standalone wins; if they are visually similar, the classification head overrides this bias to link them.

**Inference.** At test time, we discard the cooperation head and use only the classification head to predict labels in the universal taxonomy.

## 4 EXPERIMENTS

We evaluate SLAMDUNKS on the task of semantic relation discovery across datasets. To this end, we describe our experimental setup, the construction of synthetic dataset collections, evaluation metrics, baselines, and implementation details.

## 4.1 Experimental Setup

Most prior work has concentrated on semantic segmentation benchmarks. While realistic, these require very large models and long training cycles, making systematic evaluation difficult. To enable controlled and reproducible experiments, we instead focus on the image classification setting.

Specifically, we create *synthetic dataset collections* by splitting and re-labeling existing classification datasets. This approach has two advantages: (i) it allows us to define *ground-truth relations* between classes explicitly, and (ii) it provides a corresponding *universal taxonomy* against which recovered relations can be validated.

With ground truth available, we can move beyond the common—but potentially misleading—metric of per-dataset accuracy. Instead, we directly evaluate (i) the correctness of extracted relations and (ii) performance in the unified label space, which better reflects the objectives of multi-dataset training.

## 4.2 Datasets

We create 5 synthetic dataset pairs with MNIST Deng (2012), CIFAR-10 Krizhevsky (2009) and SVHN Netzer et al. (2011). All of these datasets have 10 classes in the original label space. For four of the five pairs we create two custom taxonomies by merging some classes and omitting others. We present the two splits by listing the concepts represented by their indices in the original label space. Dataset 1 has six classes: $\{0\}, \{1, 2\}, \{3\}, \{4\}, \{5, 6, 7\}$ and $\{8\}$. Dataset 2 has eight classes: $\{1\}, \{2, 3\}, \{4\}, \{5\}, \{6\}, \{7\}, \{8\}, \{9\}$. These splits cover all interesting scenarios between classes: equality between classes, subset-superset relation, overlapping classes and not related classes. We denote the custom taxonomy with the extension "-C" in the pairs name

**MNIST-C** and **CIFAR-C** are made by splitting MNIST and CIFAR10 into two subsets of equal size. These datasets represent the simplest experimental setting, as the dataset pairs share the domain.

**MNIST-SVHN** uses MNIST and SVHN as the dataset pair. These datasets share a taxonomy (digits), which means that the only relation that exists between classes is equality. Still, this setup is challenging, as there is a significant domain shift between the two datasets.

Finally, **MNIST-SVHN-C** and **SVHN-MNIST-C** apply the custom taxonomies to MNIST and SVHN, where the ordering indicates which dataset has taxonomy 1, and which taxonomy 2. This experimental setup is the most challenging as there is both a domain shift and complex relations between classes

## 4.3 Metrics

A common way to evaluate multi-dataset training is to report performance on each individual dataset. However, this can be misleading: a large model may simply memorize and separate datasets without truly sharing knowledge or discovering meaningful relations.

Since our synthetic dataset collections provide ground-truth relations and a known universal taxonomy, we evaluate *relation discovery* directly as a standalone task, alongside classification performance in the shared label space.

**Relation discovery as classification.** As described in Subsection 3.1, inter-dataset relations can be represented as a bipartite graph connecting classes across datasets. The total number of candidate relations is $(|D_1|+1)(|D_2|+1)-1$. Thus, relation discovery can be viewed as a binary classification problem, where the model must decide whether each potential relation is valid. If a method outputs continuous confidence scores for relations, we measure performance using **Average Precision (AP)**, which evaluates the precision–recall tradeoff without requiring a threshold.

**Graph-based evaluation.** If a method instead outputs a discrete set of relations, we compare the resulting bipartite graph to the ground truth using **Graph Edit Distance (GED)**. We count the number of edge additions and deletions required to transform the predicted graph into the true one.

| Method | MNIST-C AP | CIFAR-10-C AP | MNIST-SVHN AP | MNIST-SVHN-C AP | SVHN-MNIST-C AP |
|---|---|---|---|---|---|
| Missing link | $0.8 \pm 0.0$ | $0.8 \pm 0.0$ | $0.6 \pm 0.1$ | $0.6 \pm 0.1$ | $0.5 \pm 0.1$ |
| SLAMDUNKS | $1.0 \pm 0.0$ | $1.0 \pm 0.0$ | $1.0 \pm 0.0$ | $0.7 \pm 0.1$ | $0.7 \pm 0.1$ |
| Method | MNIST-C GED | CIFAR-10-C GED | MNIST-SVHN GED | MNIST-SVHN-C GED | SVHN-MNIST-C GED |
| AUT | $5.4 \pm 1.0$ | $4.2 \pm 0.4$ | $4.6 \pm 7.8$ | $6.2 \pm 4.0$ | $7.0 \pm 2.7$ |
| SLAMDUNKS | $0.0 \pm 0.0$ | $0.4 \pm 0.8$ | $4.0 \pm 2.2$ | $8.8 \pm 1.3$ | $14.4 \pm 2.6$ |

Table 1: Quantitative results of relation discovery on custom classification dataset pairs. We measure average precision in the top part of the table to compare our approach to Missing link, and graph edit distance to compare our approach to Automatic Universal Taxonomies.

**Classification accuracy.** Finally, because SLAMDUNKS simultaneously learns to classify samples into the universal taxonomy, we also report **accuracy in the shared label space**. This provides an additional measure of how well the recovered relations support effective multi-dataset training.

## 4.4 BASELINES

We compare SLAMDUNKS against two existing approaches, adapted from semantic segmentation to the image classification setting: (i) **Missing Link** Uijlings et al. (2022), and (ii) **Automatic Universal Taxonomies (AUT)** Bevandić & Šegvić (2022).

**Missing Link.** This method trains a model on dataset $A$ and applies it to dataset $B$ to estimate correlations between predictions and ground-truth labels. We extend the approach to open-set recognition by using max-softmax to detect outliers. Repeating this procedure in both directions yields two correlation matrices, which can be interpreted as weighted bipartite graphs. The final graph is obtained by merging edges and taking the larger of the two weights. Since this method does not output a conclusive relation graph, we evaluate it using **Average Precision (AP)** only.

**Automatic Universal Taxonomies (AUT).** AUT trains a model whose output label space is formed by concatenating the classes from both datasets. Two co-occurrence matrices are then computed, one for dataset $A$ and one for dataset $B$. We adapt the approach to include open-set prediction via max-softmax. Only the strongest connections are retained, followed by pruning of contradictory edges that degrade individual dataset performance. Because this method outputs only a discrete set of accepted relations, we evaluate it using **Graph Edit Distance (GED)**.

## 4.5 IMPLEMENTATION DETAILS

We use a ResNet-18 backbone He et al. (2016) initialized with ImageNet pretraining. Each head consists of a single linear layer with weights and biases initialized to zero. Training is performed with mini-batches corresponding to $0.5\%$ of the training set (but never fewer than 10 samples). Each batch is balanced to contain an equal number of samples from both datasets. To further stabilize training, we applied weighted sampling that favors rarer classes within each dataset. For MNIST and CIFAR experiments, we train for 5 epochs with a learning rate of $1 \times 10^{-4}$ with Adam optimizer. For MNIST–SVHN experiments, we train for 20 epochs with a learning rate of $5 \times 10^{-4}$ and resize the images to 32 $x$ 32 pixels. We do not apply weight decay. The coefficient $\lambda$ in Equation 8 is selected via a simple linear search in $[0, 1]$. All reported results are the mean and variance over five independent runs. Experiments are conducted on a single NVIDIA GTX 1080 GPU.

## 5 RESULTS

We organize our results into three parts. First, we evaluate the relation discovery ability of SLAM-DUNKS by comparing it with baseline methods. Second, we present ablation studies that assess the impact of key design choices. Finally, we analyze the feature spaces learned by different multi-dataset training methods.

| pretrain | zero init | weighted sampling | MNIST-C | | CIFAR-10-C | | MNIST-SVHN | |
|---|---|---|---|---|---|---|---|---|
| | | | AP | GED | AP | GED | AP | GED |
| | | | $0.6 \pm 0.2$ | $7.6 \pm 4.3$ | $0.3 \pm 0.1$ | $13.4 \pm 2.7$ | $0.1 \pm 0.0$ | $26.6 \pm 2.1$ |
| ✓ | | | $0.4 \pm 0.1$ | $11.4 \pm 3.8$ | $0.3 \pm 0.1$ | $12.8 \pm 3.2$ | $0.1 \pm 0.0$ | $28.8 \pm 2.0$ |
| | ✓ | | $0.8 \pm 0.0$ | $4.0 \pm 0.0$ | $0.8 \pm 0.1$ | $6.0 \pm 0.0$ | $0.6 \pm 0.2$ | $16.6 \pm 3.1$ |
| ✓ | ✓ | | $1.0 \pm 0.0$ | $0.0 \pm 0.0$ | $0.8 \pm 0.2$ | $2.0 \pm 2.4$ | $0.8 \pm 0.1$ | $11.0 \pm 3.0$ |
| | | ✓ | $0.7 \pm 0.1$ | $5.6 \pm 2.7$ | $0.6 \pm 0.3$ | $10.0 \pm 6.1$ | $0.2 \pm 0.1$ | $25.8 \pm 2.1$ |
| ✓ | | ✓ | $0.5 \pm 0.2$ | $10.8 \pm 2.8$ | $0.2 \pm 0.1$ | $16.6 \pm 2.2$ | $0.1 \pm 0.1$ | $28.8 \pm 2.1$ |
| | ✓ | ✓ | $0.7 \pm 0.1$ | $4.0 \pm 0.0$ | $0.7 \pm 0.0$ | $6.0 \pm 0.0$ | $0.6 \pm 0.2$ | $12.2 \pm 6.0$ |
| ✓ | ✓ | ✓ | $1.0 \pm 0.0$ | $0.0 \pm 0.0$ | $1.0 \pm 0.0$ | $0.4 \pm 0.8$ | $1.0 \pm 0.0$ | $4.0 \pm 2.2$ |

Table 2: Quantitative results on custom classification dataset pairs

## 5.1 RELATION RECOVERY

Our first set of experiments compares the proposed *SLAMDUNKS* with two baselines: (i) Missing Link Uijlings et al. (2022), and (ii) Automatic Universal Taxonomies Bevandić & Šegvić (2022).

For Missing Link, we report only AP, since producing a conclusive relation graph would require thresholding its outputs. For AUT, we report only GED, as the method does not yield confidence scores for relations. Table 1 summarizes the results.

SLAMDUNKS clearly outperforms Missing Link across all dataset pairs. Missing Link relies on training two separate models, each of which tends to overfit to its respective dataset. This limits cross-dataset generalization and produces inferior correlation matrices for relation discovery.

On same-domain dataset pairs, SLAMDUNKS also surpasses AUT. AUT frequently introduces spurious relations for outlier classes, despite having a pruning step to mitigate them. On cross-domain dataset pairs, results are more mixed: while AUT continues to suffer from spurious connections, SLAMDUNKS sometimes separates classes that should be connected. This behavior stems from the gating head, which more readily favors standalone assignments under domain shift.

## 5.2 ABLATION STUDIES

We next investigate how different design choices affect the performance of *SLAMDUNKS*. Our analysis covers initialization strategies, sampling, pretraining, parameter $\lambda$, and dataset size.

Two considerations are particularly important: (i) initializing the weights and biases of both heads to zero, and (ii) applying weighted sampling across classes. Zero initialization avoids unintended bias at the start of training: in the classification head, all potential relations are equally likely, and in the gating head, each class is equally likely to be shared or standalone. Weighted sampling reduces the natural tendency of the gating head to default to predicting frequent classes as standalone.

We also study the effect of initializing the backbone with pretrained ImageNet features. Pretraining provides limited benefit on its own, but it improves performance when combined with zero initialization and weighted sampling. This suggests that pretrained features alone do not capture the semantics required for aligning heterogeneous datasets.

Our approach combines two losses 8 and balanced by the parameter $\lambda$. Figure 3 shows the effect of varying $\lambda$ in terms of AP, GED, and classification accuracy. For same-domain datasets, higher values of $\lambda$ yield the best results, as strong visual similarity makes it easy for the model to connect classes. For cross-domain datasets, lower $\lambda$ performs better, as domain shift makes it easier for the

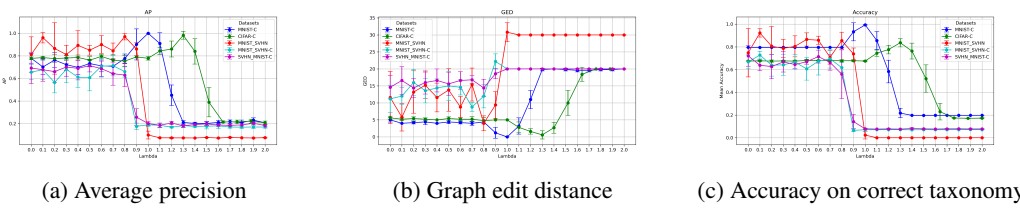

(a) Average precision     (b) Graph edit distance     (c) Accuracy on correct taxonomy

Figure 3: Impact of lambda on taxonomy construction quality

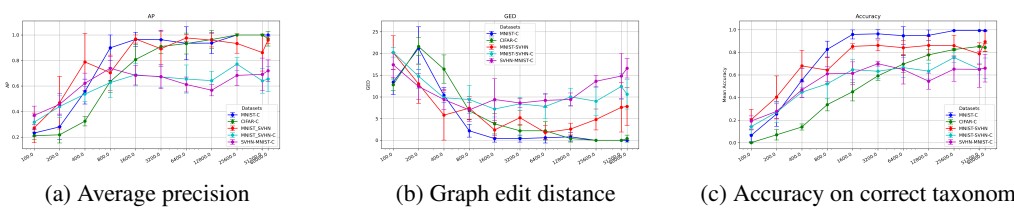

(a) Average precision      (b) Graph edit distance      (c) Accuracy on correct taxonomy

Figure 4: Impact of training set size on taxonomy construction quality

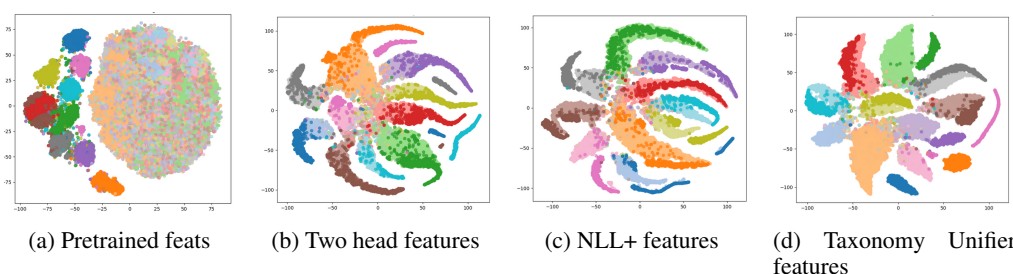

(a) Pretrained feats      (b) Two head features      (c) NLL+ features      (d) Taxonomy Unifier features

Figure 5: Visualization of feature spaces for SVHN and MNIST sammple embeddings for different models

gating head to treat each class as standalone. Interestingly, classification accuracy does not always correspond to GED, indicating incorrect relations may affect relatively few samples in practice.

Finally, we analyze how dataset size influences performance (Figure 4). Larger datasets consistently improve relation discovery, but even with only 25% of the training data, the model already achieves strong results. This suggests that reliable relation discovery can be obtained using subsets of available data, offering a path toward more efficient training.

## 5.3 MULTI-DATASET TRAINING

We next analyze how different training strategies affect the latent space learned by the feature extractor. We compare SLAMDUNKS against: (i) a pretrained model, (ii) training with dataset-specific heads Kalluri et al. (2019), and (iii) the NLL+ loss Bevandić et al. (2024). All experiments are conducted on MNIST and SVHN, and the resulting feature spaces are visualized in Figure 5.

Using a pretrained ResNet-18 backbone without further adaptation produces feature spaces that clearly separate MNIST from SVHN, with no semantic alignment between related classes. Training with NLL+ in the known universal label space achieves stronger alignment: universal classes are cleanly separated, and samples from both datasets are well mixed within each class. Our approach seems to further improve representation quality. It aligns semantically related classes across datasets, while at the same time preserving a distinction between domains. This demonstrates that SLAMDUNKS simultaneously supports cross-dataset semantic consistency and domain awareness.

## 6 CONCLUSION

We addressed multi-dataset training and relation discovery in settings where datasets share concepts but have unaligned taxonomies. Our proposed SLAMDUNKS learns both classification and class relations through two competing heads on a shared backbone: one predicting whether classes are standalone, and the other mapping samples into a universal taxonomy. This design enables discovery of shared concepts while preserving standalone categories.

Using synthetic dataset pairs with controllable ground-truth relations, we showed that our method outperforms prior approaches, particularly in handling standalone classes, and yields more informative feature spaces. Future work will extend this framework to more complex tasks.

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
