# OpenReview forum: "Finding agreement in disagreement: Simultaneous label alignment and multi-dataset training with SLAMDUNKS"
_ICLR.cc/2026/Conference — ICLR 2026 Conference Withdrawn Submission_

### Official Review · Reviewer_X1V5 · 2025-10-21

**Soundness:** 2
**Presentation:** 2
**Contribution:** 2
**Rating:** 2
**Confidence:** 4

**Summary:**

This paper identifies the challenge of inconsistent and contradictory taxonomies as a major obstacle in multi-dataset training, which introduces noise and limits knowledge sharing. They proposes SLAMDUNKS, a novel framework that performs simultaneous multi-dataset training and label alignment through a shared feature extractor with dual competing heads—a gating head for selecting shareable dataset-specific classes and a classification head for mapping samples to a unified taxonomy.

**Strengths:**

1. The paper has a generally complete structure and presents its content clearly.

2. The motivation of the article is clear.

3. The research problem addressed in the paper is quite interesting.

**Weaknesses:**

1. The classification method implemented in this paper is based on an exhaustive approach, with the probability derivation being quite basic. Additionally, the design of the loss function is mostly heuristic, lacking theoretical guarantees.

2. The multi-dataset training experiments were conducted only in the context of two datasets, with no attempts made to explore three or more datasets. Furthermore, there is no targeted design in the methodology, which undermines the credibility of both the experimental results and the methods presented in the paper.

3. The writing lacks rigor, and many symbols are not clearly defined.

4. The diagrams and tables are not sufficiently clear, and the tables lack outer borders.

**Questions:**

The current version of the paper is significantly far from being ready for publication. The authors are encouraged to address the aforementioned shortcomings effectively.

---

### Official Review · Reviewer_CYiH · 2025-10-31

**Soundness:** 3
**Presentation:** 3
**Contribution:** 2
**Rating:** 2
**Confidence:** 4

**Summary:**

The paper notes that multi-dataset training is hindered by unaligned/contradictory taxonomies. It proposes SLAMDUNKS, a framework with a shared feature extractor and two competing heads (gating, classification) for training and label alignment. A synthetic benchmark is introduced. Results show perfect same-domain taxonomy recovery (GED 0) and better cross-domain AP (0.8) than state-of-the-art.

**Strengths:**

The results of the experiment are promising.

**Weaknesses:**

1. The theoretical analysis of the dual-head interaction lacks sufficient depth. The paper selects the hyperparameter λ (controlling the standalone bias in the loss function, Eq. 8) via linear search in [0,1], but it does not explain the theoretical basis for restricting λ to this range. Given that λ’s optimal value varies across datasets (e.g., higher λ for same-domain pairs and lower λ for cross-domain pairs, Fig. 3), there is no discussion of how dataset characteristics (e.g., domain shift magnitude, class overlap ratio) quantitatively influence λ’s optimal value.
2. The paper only validates the framework on image classification tasks, despite mentioning in the introduction that multi-dataset training is critical for semantic segmentation, medicine, and climate science. The absence of experiments on semantic segmentation (a task where label taxonomy inconsistencies are more common, e.g., COCO vs. ADE20K) makes it unclear whether SLAMDUNKS can generalize beyond classification.
3. The analysis of ablation results lacks depth and completeness. The paper claims that zero initialization of the heads "avoids unintended bias," but it does not compare zero initialization with other common initialization methods. Without this comparison, it is impossible to confirm whether "zero initialization is optimal" is a universal conclusion or a task-specific result.
4. The analysis of results is incomplete, and practical application considerations are insufficient. The paper mentions in Section 5.1 that SLAMDUNKS "sometimes separates classes that should be connected" in cross-domain scenarios, but it does not analyze the root causes of these errors. Moreover, the paper only evaluates two-dataset scenarios, but practical multi-dataset training often involves three or more datasets. The paper does not discuss how SLAMDUNKS would scale to multi-dataset settings.

**Questions:**

See Weaknesses.

---

### Official Review · Reviewer_beds · 2025-11-01

**Soundness:** 3
**Presentation:** 2
**Contribution:** 1
**Rating:** 2
**Confidence:** 2

**Summary:**

The paper tackles the problem of inconsistent dataset taxonomies and prevents using multi-dataset combining in training. The authors propose training a shared feature extractor with two competing heads, a gating head to determine dataset-specific classes and a classification head to map samples to shared taxonomy emerging from training.

The authors perform some evaluation using a synthetic benchmark. They demonstrate validation on a few small-scale classification tasks. The methods provide a comprehensive discussion to a few related work, e.g. DynAlign and LMSeg, but it is yet clear how they would differentiate in the proposed settings and evaluations. This is part the method can potentially be further improved.

**Strengths:**

SLAMDUNKS introduces a novel approach that unifies label alignment and multi-dataset training within a single end-to-end model. This design allows automatic discovery of semantic relations across datasets without manual mappings or reliance on external language models.

The paper also creates a synthetic benchmark with ground-truth taxonomies. It allows the paper and future work to define ground truth relations explicitly and a taxonomy setting which be validated. Under this setting, the methods can outperform existig baselines Missing Link and AUT.

**Weaknesses:**

Although the small-scale experiments on synthetic dataset is clean and nice, it was not intuitive to me how the approach would extend to real-world tasks such as segmentation or detection. There are plenty existing classic datasets in this spaces. Making some evaluation with efforts constructing a similar setting will be impactful.

In addition, while the paper shows clear advantage over the two classical baselines, both baselines Missing Link and AUT are not frontier according to the discussions in the related work. It would be good if the authors can bringer the mentioned newer work such as DynAlign, LMSeg in the evaluation loop, which would be enhance the claim to outperform state-of-the-art.

**Questions:**

I am overall less familiar with the literature in this space, so I raise the questions whether there are any improvements the authors can make to push this study closer to real world scenarios and validate with more recent works. Please help me clarify if there are clear distinctions that the suggestions are not appropriate. I will also be open to other reviewers' suggestion on this.

---

### Official Review · Reviewer_wkJ8 · 2025-11-01

**Soundness:** 2
**Presentation:** 2
**Contribution:** 1
**Rating:** 2
**Confidence:** 4

**Summary:**

The author proposed a framework for simultaneous multi-dataset training and label alignment and a synthetic benchmark to address the problem of the inconsistencies across the datasets.

**Strengths:**

1. The idea of addressing the inconsistencies problem is intereseting and practical.

**Weaknesses:**

1. The paper is hard to follow. The orgnaization of the paper should be improved. The Figure is not well drawen, and it is hard to interpret. The mathmatical formulations are not formal.
2. Since the gating after the sigmod would result in non-zero logits, how the author choose the standlone or not? By the value? The author should clarify this point.
3. Since you mentioned the relationship discovery between two datasets, I see no advantage in your paper can guarantee the quality of relationship discivery.
4. The method and the loss in the paper is not reasonable and the method is not well-motivated by your motivation.
5. For the Equation 6, since it is a probability, the design of this equation is not good as it may exceed 1.

**Questions:**

See weakness

---

### Note · Authors · 2025-12-03

I have read and agree with the venue's withdrawal policy on behalf of myself and my co-authors.